# Spatial Distribution of Canine Sporotrichosis in Rio de Janeiro, Brazil (1998–2018) and Its Correlation with Socioeconomic Conditions

**DOI:** 10.3390/jof8111207

**Published:** 2022-11-15

**Authors:** Anna Barreto Fernandes Figueiredo, Mônica de Avelar Figueiredo Mafra Magalhães, Wagner de Souza Tassinari, Isabella Dib Ferreira Gremião, Luisa Helena Monteiro de Miranda, Rodrigo Caldas Menezes, Sandro Antonio Pereira

**Affiliations:** 1Laboratory of Clinical Research on Dermatozoonoses in Domestic Animals, Evandro Chagas National Institute of Infectious Diseases, Oswaldo Cruz Foundation, Rio de Janeiro 21040-900, Brazil; 2GIS Laboratory, Health Information Laboratory, Institute of Scientific and Technological Information and Communication in Health, Oswaldo Cruz Foundation, Rio de Janeiro 21040-900, Brazil; 3Mathematics Department, Exact Sciences Institute, Federal Rural University of Rio de Janeiro, Rio de Janeiro 23897-000, Brazil; 4Veterinary Pathology Diagnostic Services, Sydney School of Veterinary Science, The University of Sydney, Sydney 2006, Australia

**Keywords:** cat, sporotrichosis, epidemiology, spatial analysis

## Abstract

Canine sporotrichosis is a poorly described global disease and a spatial approach has not yet been used to assess the disease in dogs. Therefore, this study aimed to describe the occurrence of canine sporotrichosis in the state of Rio de Janeiro, Brazil, from 1998 to 2018 and its correlation with socioeconomic characteristics using exploratory spatial data analysis. A total of 295 cases of canine sporotrichosis were identified and 249 were georeferenced. There was a higher concentration of cases in the municipality of Rio de Janeiro, as well as along the border of the city and the adjacent municipalities in the great metropolitan area. The cases occurred in areas where most of the dwellings are houses. Moreover, no focus of disease density was found in the southern part of Rio de Janeiro city over the period studied, possibly due to better socioeconomic conditions. Areas with a high concentration of canine sporotrichosis cases coincided with regions that possessed a low proportion of households without paving, suggesting that the disease is not necessarily linked to extreme poverty. The mapping of areas with a greater density of cases is fundamental to formulate targeted and strategic plans in order to implement effective public health prevention and control measures.

## 1. Introduction

Sporotrichosis is a subcutaneous mycosis caused by species of the pathogenic clade of the genus *Sporothrix* (*S. brasiliensis*, *S. schenckii*, *S. globosa*, and *S. luriei*), which affects humans and animal species, mainly cats and dogs [1,2]. The disease has a worldwide distribution, with a high prevalence in tropical and subtropical areas, especially in the Americas and Asia [3,4,5]. Since 1998, Brazil has experienced geographic expansion of sporotrichosis, particularly in the southern and southeastern regions, with a large number of cases in humans, dogs, and cats [6]. The metropolitan region (MR) of Rio de Janeiro, located in the southeastern region of Brazil, is the main endemic area. Further, an increasing frequency of cat-transmitted sporotrichosis has been observed since the 1990s [7,8]. Among the causative agents of sporotrichosis, *S. brasiliensis* has been described as the most virulent species, with a predominant geographic distribution in Brazil [2,6,9,10].

Canine sporotrichosis is a poorly described global disease. The first cases were reported in France and in the United States at the beginning of the last century [11,12]. In Brazil, the first two canine cases were only described in the 1950s [13]. Other cases have been reported in Italy [14] and in the United States [15]. Today, Brazil has the largest number of reports of canine sporotrichosis, with most published cases occurring in the MR of Rio de Janeiro [1,9,16,17]. Dogs typically become infected during hunting activities, where possible transmission of *Sporothrix* sp. can be achieved through thorns or wood splinters [18]. However, in Rio de Janeiro, specifically, dogs are mainly infected by close contact with infected cats [1]. 

The most frequent form of sporotrichosis reported in dogs is the cutaneous form, which is usually characterized by ulcers and/or nodules on the head, ears, neck, and thorax [9,19]. Nasal mucosa lesions are common and are often accompanied by extracutaneous signs, particularly respiratory signs (i.e., sneezing, dyspnea, and nasal discharge) [20]. Most canine cases exhibit lymph node enlargement; however, it must be noted that lymphangitis is uncommon [1,15]. Furthermore, osteoarticular involvement and disseminated forms are rarely observed [15,21,22]. 

Few cases of canine sporotrichosis have been documented; additionally, a spatial approach has scarcely been used to assess the disease in dogs [23]. Therefore, this study aimed to describe the occurrences of canine sporotrichosis in the state of Rio de Janeiro, Brazil, from 1998 to 2018 and its correlation with the socioeconomic characteristics of the owners using techniques of exploratory spatial data analysis.

As canine sporotrichosis is usually associated with feline sporotrichosis which, in turn, is associated with human sporotrichosis (in the context of zoonotic transmission) these results could help support the decision-making of the relevant health authorities.

## 2. Materials and Methods

This was an analytic ecological study of canine sporotrichosis conducted at the Laboratório de Pesquisa Clínica em Dermatozoonoses em Animais Domésticos (Lapclin-Dermzoo), Instituto Nacional de Infectologia Evandro Chagas (INI)/Fiocruz, between 1998 and 2018. The Lapclin-Dermzoo is a reference center for the diagnosis and treatment of sporotrichosis in animals within the state of Rio de Janeiro. This state is located in the southeastern region of Brazil, has a population of 15,989,929 inhabitants, and occupies an area of 43,780.157 km^2^ [24].

The 92 municipalities are divided into eight regions, with the: Metropolitan region, Coastal Shore region, Northern Fluminense region, Northwest Fluminense region, Mountain region, Central South Fluminense region, Middle Paraíba region, and Green Coast region. In regard to 18 of these municipalities, the MR has the second largest urban population in the country. According to the Regionalization Master Plan, the MR is divided into two poles on opposite sides of the Guanabara Bay: MR I with 11 municipalities (Rio de Janeiro, Belford Roxo, Duque de Caxias, Japeri, Magé, Mesquita, Nilópolis, Nova Iguaçu, Queimados, São João de Meriti, and Seropédica) and MR II with seven municipalities (Itaboraí, Maricá, Niterói, Rio Bonito, São Gonçalo, Silva Jardim, and Tanguá) [25]. 

The majority (73.9%) of the state’s population lives in MR I and II. MR I is the most populous (61.7%), with the number of residents being five times larger than that of MR II (12.2%) [25]. 

The database of Lapclin-Dermzoo, INI/Fiocruz, comprising the period from 1998 to 2018, was used as the data source in this study. Each animal had an individual medical record number provided at the first visit. This registration number was valid for all subsequent consultations at INI/Fiocruz throughout the patient’s life. All dogs were diagnosed via isolation and the identification of *Sporothrix* sp. in appropriate culture media [1], which is the reference method for the diagnosis of sporotrichosis. The T3B PCR fingerprinting technique was used to characterize the isolates of *Sporothrix* sp. at the species level [10,26].

According to the location of the lesions, clinical forms were classified into cutaneous (localized/fixed, lymphocutaneous, or disseminated cutaneous form) and mucosal forms. The animal could present only one clinical form or more, regardless of the presence or absence of extracutaneous signs (i.e., lymphadenopathy or respiratory signs) [17,20] (Figure 1).

The cartographic base used was the digital grid of the state of Rio de Janeiro, which is available in shape format (i.e., a vector format of geospatial data). Further, this digital map is subdivided into municipalities and census tracts. The map was provided by the Instituto Brasileiro de Geografia e Estatística (IBGE). The data on the resident population in each census tract, as well as the socioeconomic variables, were taken from the 2010 Demographic Census [24], which was released in a digital format by IBGE.

The variables served as a basis to build the indicators used in the data analysis, that is the: proportion of houses as dwelling type (P_DOM_CAS); proportion of literate responsible persons (P_R_ALF); proportion of households with sewage via a rudimentary cesspool or ditch (P_DOM_ESG); proportion of households with garbage thrown in the river, lake, or sea (P_DOM_LIX); proportion of responsible persons with a monthly income of up to 1 minimum wage (P_R_1SM); proportion of households without paving (P_DOM_SPA); proportion of households with garbage accumulated in the street (P_DOM_LLO). Those indicators reflect income, schooling, housing, and sanitation conditions, thereby estimating the population’s living conditions. 

The addresses of canine cases were georeferenced with the residence address in three stages, which are: (I) the manual standardization of addresses, (II) georeferencing using Google Maps, and (III) manual intervention. The standardization of addresses sought to correct frequent errors, to remove strange characters, and to standardize components related to the type (avenue, square, street, etc.) and title (for example, President, Teacher, Princess, etc.) of public places. Although this step did not eliminate all errors, it was very useful for the purposes of increasing the number of automatically georeferenced addresses.

Georeferencing was performed using the Google Application Programming Interface (API), accessed with a script programmed in R version 4.0.5. The API compares the addresses with the database of the programs in order to capture geographic coordinates. The API also returns the localized address and its accuracy. To verify the quality of the georeferencing, the addresses returned were compared with those reported. In the case of disagreement between addresses, they were transferred to the manual intervention together with the other missing addresses. In addition, a sample of georeferenced addresses was manually analyzed in order to determine whether the coordinate corresponded to the address provided.

In the manual step, the remaining spelling errors were corrected, and the required manual searches were conducted. The addresses of the dogs’ residences in this manual step were georeferenced using Bing Maps. 

The geographical coordinates were plotted as points for the purposes of mathematical operations between layers and for preparation of the thematic maps. The thematic maps were constructed by a geoprocessing technique using QGis 3.8.1 (Open Source Geospatial Foundation) and ArcGis 10.2 (ESRI, 2011) software.

The canine sporotrichosis cases were divided according to certain three-year intervals (i.e, 1998–2000, 2001–2003, 2004–2006, 2007–2009, 2010–2012, 2013–2015, and 2016–2018). The cases were then implemented into files that originated kernel maps for each period, as well as for the total period (1998–2018). 

For the analysis of sociodemographic indicators, the census sector was chosen as a spatial unit, which is a territorial unit created by IBGE for the organization and execution of censuses and research. The census sector is the smallest unit of aggregation in which the data collected in the Demographic Censuses are released.

We analyzed the hot areas from kernel maps of the entire period, according to sociodemographic indicators. For this purpose, the points were categorized into three density strata and classified into high density, medium density, and low density. The construction of the kernel map permitted the identification of areas with the highest density of cases and the function “Sample values of Raster” of QGis allowed us to identify in which density stratum each point was located. 

In addition to exploratory analysis, the nonparametric Kruskal–Wallis test was used in order to identify possible differences between socioeconomic indicators and the density of cases. In addition, the Kendall rank correlation test was applied to investigate possible associations between case density groups and socioeconomic indicators [27]. A level of significance of 5% was adopted for all statistical tests. Further, statistical analysis was performed using the R software, version 4.0.5.

## 3. Results

During the period of the study (1998–2018), 295 cases of canine sporotrichosis were identified. Most dogs were male (61.1%) and the median age was 50.5 months. The disease affected different breeds, with a predominance of mongrel dogs (48.6%), followed by poodle (14.8%), pinscher (7.9%), and then pitbull (4.2%), among others.

The predominant clinical presentation was the mucosal form (26.1%) followed by the localized/fixed form (25.3%), disseminated cutaneous form (17.8%), localized/fixed form associated with the mucosal form (15.8%), disseminated cutaneous form associated with the mucosal form (11.9%), and then the lymphocutaneous form (3.1%). Respiratory signs were observed in 62.3% of the dogs, and rhinorrhea was the most common instance of them, which was then followed by sneezing and dyspnea. 

Due to the retrospective design of this study, not all data were available for each animal. In regard to the variables (i.e., gender, age, breed, and the clinical presentation), 2, 39, 32, and 3 of the 295 cases were excluded, respectively, due to missing data.

Of the 295 cases of canine sporotrichosis identified in the study, 46 isolates were previously characterized as *S. brasiliensis*, and one isolate was characterized as *S. schenckii* [9].

Figure 2 shows the distribution of canine sporotrichosis cases in the state of Rio de Janeiro between 1998 and 2018. There was a higher concentration of cases in the municipality of Rio de Janeiro and along the border between this city and the adjacent municipalities in the greater metropolitan area. 

Analysis of the distribution of the disease according to the municipality of residence (Table 1) allowed us to identify cases of canine sporotrichosis in 17 of the 92 municipalities (18%) in the state of Rio de Janeiro. In addition, 97.14% of the cases occurred in patients residing in 11 municipalities of MR I. In MR II, only three municipalities had cases of the disease and the proportion of cases was also much lower than that observed in MR I (1.78%). The Mountain and Coastal Shore regions accounted for 0.36% and 0.72% of the cases, respectively. No cases of canine sporotrichosis were observed in the other regions of the state during the study period. The municipality of Rio Janeiro accounted for 54.28% of all cases of canine sporotrichosis in the state, followed by the municipalities of Duque de Caxias (22.86%), São João de Meriti (9.28%), and Nova Iguaçu (5.36%).

Of the 295 records of dogs, 36 did not have a full address or, instead, the information was limited (e.g., only the neighborhood or municipality was known). Of the remaining 259 records submitted to the procedure, 237 (80.34%) were automatically georeferenced. The agreement between the addresses informed the analysis and those found were 100% in the manually analyzed sample (24 addresses). The remaining 22 addresses were referred for manual intervention, which recovered 12 (54.55%) of the addresses not initially located. Finally, 249 were georeferenced.

Figure 3 shows the kernel maps of canine sporotrichosis between 1998 and 2018, divided into three-year intervals (1998–2000, 2001–2003, 2004–2006, 2007–2009, 2010–2012, 2013–2015, and 2016–2018). A first visual analysis shows that the “hot” areas with the highest density of canine sporotrichosis cases expanded geographically over the studied period, with the spatial pattern of cases persisting over the years.

Figure 4 shows the kernel map of canine sporotrichosis for the georeferenced points over the total period from 1998 to 2018, highlighting the locations of greater occurrence.

Figure 5 shows the cases of canine sporotrichosis in the state of Rio de Janeiro between 1998 and 2018, superimposed on the proportions of socioeconomic indicators with some distribution pattern observed for the variables. The first map (A) shows that areas with a high concentration of canine sporotrichosis cases coincided with regions with a high proportion of houses as dwelling type (P_DOM_CAS). In the case of the proportion of literate responsible persons (P_R_ALF), practically the entire region where canine sporotrichosis cases were found had a high proportion of this variable (B). The proportion of households with garbage thrown in the river, lake, or sea was low in the region studied (P_DOM_LIX) (C). Areas with a high concentration of canine sporotrichosis cases coincided with regions with a low proportion of households without paving (P_DOM_SPA) (D). The socioeconomic indicators P_R_1SM, P_DOM_LLO, and P_DOM_ESG had no distribution pattern visually observed. 

According to the results gained from the application of the Kruskal–Wallis test (Table 2), only the proportion of households without paving (P_DOM_SPA), among all social indicators, showed a significantly different distribution amongst case densities: low (average 0.166, range 0.000–1.000); medium (average 0.059, range 0.000–0.523); and high (average 0.056, range 0.000–0.980). 

Regarding the proportion of households with garbage accumulated in the street (P_DOM_LLO), this characteristic was more frequently found in the census sectors classified as high density of cases, although the difference was not statistically significant.

With respect to the proportion of households with garbage thrown in the river, lake or sea (P_DOM_LIX), this proportion was zero (0%) in sectors classified as low density and medium density, indicating that only sectors classified as high density contained households in these conditions.

According to the Kendall coefficient that indicates the correlation between case density and social indicators at the census sector level, a significant negative correlation at 5% (−0.164) was observed for the proportion of households without paving (P_DOM_SPA), thereby corroborating the findings of the previous analysis.

## 4. Discussion

Since the late 1990s, few studies have investigated canine sporotrichosis (most of them were case reports or case series). Generally, these articles focused on clinical and therapeutic features, while epidemiological data were poorly explored. Thus, little is known about the geographical distribution of canine cases over time in the state of Rio de Janeiro, the region that has the greatest number of animal sporotrichosis cases described in the world.

To our knowledge, only one article addressed the spatial distribution of 23 canine cases in the municipality of Campos dos Goytacazes, located in the North Fluminense region in the state of Rio de Janeiro, during the period from 2016 to 2019 [23]. However, our study did not include cases from the northern region of Rio de Janeiro, due to the distance between this region and INI/Fiocruz (276 km).

In Brazil, the main etiological agent of canine sporotrichosis is *S. brasiliensis* and sporadic cases are caused by *S. schenckii* and *S. luriei* [9]. In other countries, there are no studies on the molecular identification of species that causes sporotrichosis in dogs [28].

High rates of human, feline, and canine cases of sporotrichosis caused by *S. brasiliensis* and transmitted by cats have been reported in Brazil since 1998; further, most of them have been occurring in the city of Rio de Janeiro and surroundings. The current study analyzed the geographic distribution of canine sporotrichosis over time in the state of Rio de Janeiro between 1998 and 2018 using data from dog follow-ups at INI/Fiocruz. 

The predominance of sporotrichosis in adult male mongrel dogs agrees with previously published data on canine sporotrichosis in the state of Rio de Janeiro [1]. Similar to human and feline sporotrichosis, canine sporotrichosis showed a tendency to expand in the state of Rio de Janeiro, with spatial coverage increasing over the years since 1998. According to the data described herein, the spatial distribution of canine sporotrichosis in Rio de Janeiro between 1998 and 2018 was characterized by a concentration of cases in the municipality of Rio de Janeiro and along the border, i.e., between the city of Rio de Janeiro and the adjacent municipalities in MR I. Moreover, this is considered the region with the greatest population and with the oldest urban occupation [29]. 

The occurrence of canine cases in the state of Rio de Janeiro showed a concomitant spatial expansion with human cases [7]. This fact reflects the association of canine sporotrichosis with feline sporotrichosis which, in turn, is associated with human sporotrichosis in the context of zoonotic transmission, which is extremely important in the epidemiological chain of sporotrichosis in the state of Rio de Janeiro. In the study of Barros et al. [30], 90.7% of the human patients with sporotrichosis in the same state were in domiciliary or professional contact with cats that possessed this disease.

Sporotrichosis is a neglected disease that has become a public health problem due to its considerable increase in feline cases and, consequently, in human and canine cases since the onset of the epidemic, particularly in the MR of Rio de Janeiro. These data are compatible with the hyperendemicity of human and feline sporotrichosis in the state of Rio de Janeiro. Furthermore, these cases are diagnosed at INI/Fiocruz as a result of transmission involving sick cats [31,32,33].

In the present study, the cases occurred in areas where most of the dwellings were houses. In economically underprivileged regions of the state’s capital and in the municipalities of MR, cats mostly live in houses and circulate in the neighborhood, often becoming involved in fights with other animals [7,34]. In this scenario, cats, humans, and dogs may become infected through contact with outdoor sick cats, a fact that may explain the increased number of human and animal sporotrichosis cases in the poorer areas of the MR of Rio de Janeiro [7]. 

No concentration of the disease was found in the southern area of Rio de Janeiro city over the period studied, possibly due to its better socioeconomic conditions and infrastructure, which is evidenced in the fact that it possesses the highest per capita income of the municipality [29]. In that part of the city, cats and dogs usually stay indoors because they reside in apartments; moreover, dogs are mostly kept on a leash during walks, preventing the transmission of *Sporothrix* sp. and other diseases.

Regarding the educational level of the responsible persons, sporotrichosis was not found to be a disease related to schooling. This indicator was restricted to literate/illiterate people and impaired the analysis, since this variable, together with income, is a good estimator of the population’s living conditions.

The variable P_DOM_SPA showed an opposite correlation to what was expected, both in visual analysis and in the statistical tests, whereby it was thought to represent a protective factor, i.e., the lower the proportion of households without paving, the greater the chance of canine sporotrichosis cases in these areas. This shows a change in the sporotrichosis paradigm. The disease is not necessarily linked to extreme poverty, nor does it occur only in areas with a dirty floor, but is also present in more urban and structured areas. These results corroborate the findings of a cross-sectional study that involved patients with sporotrichosis from Rio de Janeiro [35]. These authors [35] found that most dwellings were houses, with 83% of them having full access to basic sanitation. Furthermore, the streets were paved with asphalt in 75.4% of cases or with paving stone in 12.3%. In addition, it is important to highlight that people living in the most impoverished areas face a range of barriers that reduce their ability to access adequate treatment at a referral center. Therefore, some cases may not have been detected, possibly influencing the results of the study.

The need for spending on urban transport and the difficulty of transporting dogs in public transport on a periodic basis are important obstacles for individuals of lower socioeconomic status to access veterinary care and medicines. The sample studied consisted of animals whose owners were able to overcome these obstacles because of their economic status. This fact may represent a selection bias, favoring individuals with some purchasing power in these regions and, therefore, making it difficult to detect cases from municipalities far from the MR.

Another limitation of this study was that cases diagnosed at other public or private institutions and veterinary clinics could not be included. It is, therefore, difficult to comprehensively assess the occurrence and distribution of canine sporotrichosis in the state of Rio de Janeiro.

Although our study does not cover other public sporotrichosis care units or private practices, it must be noted that this demand was not so expressive in other locations, especially between 1998 and 2013. It is also observed that, over the years, INI/Fiocruz became a referral center for the cases from other institutions. Furthermore, since 2014, free-of-charge diagnosis and treatment of animals have both become available in two other public institutions in the city of Rio de Janeiro. Until then, INI/Fiocruz was responsible for most reports of animal disease in the state [36].

Since the late 1990s, human and feline sporotrichosis have been investigated in different studies but few have addressed the disease in dogs. The scarcity of published canine cases makes the investigation of these cases relevant, as canine cases are commonly linked to feline and human cases. 

Despite the important increase in cases of zoonotic transmission involving sick cats in Brazil in the last two decades, especially in Rio de Janeiro [7,30,32,33,37], there is no sufficiently robust evidence that dogs facilitate transmission of *Sporothrix* spp. to humans or other animals. However, considering the particularities of the most affected regions, the discussion becomes relevant, as it allows a real spatial view of the disease and helps in disease prevention measures and in the management of control programs. Additional epidemiological, social, and environmental studies that include more representative samples are needed to better understand this issue.

## 5. Conclusions

To our knowledge, this is the first study addressing canine sporotrichosis by mapping cases and using a sociodemographic approach. The canine cases occurred in areas where most of the dwellings are houses and showed a concomitant spatial expansion with human cases. Additionally, the results suggested that the disease is not necessarily linked to extreme poverty. These findings could be an advantage in aiding the decision-making of the relevant authorities. Future studies should involve larger samples, as well as analysis of the other regions of Brazil.

## Figures and Tables

**Figure 1 jof-08-01207-f001:**
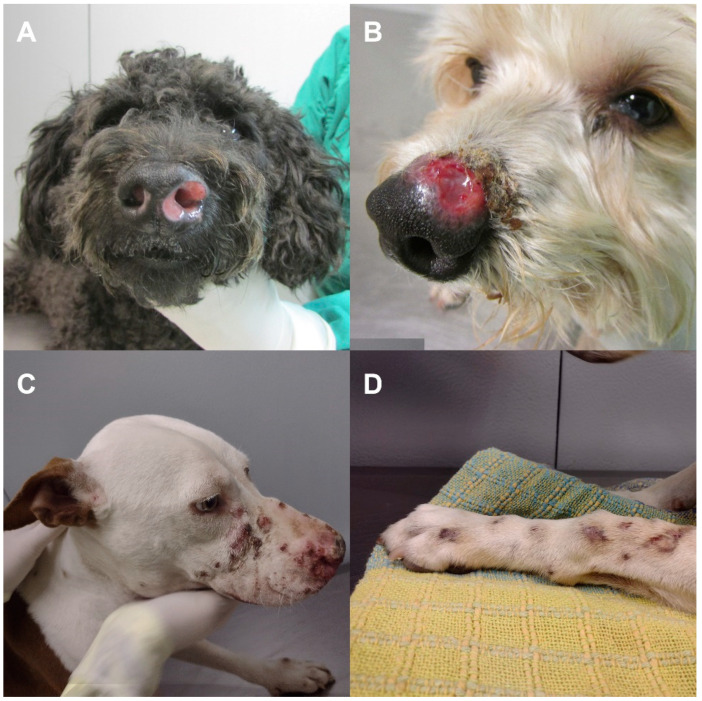
Clinical presentations of canine sporotrichosis. (**A**) Nasal mucosal lesions (mucosal form); (**B**) ulcer and meliceric crusts on the nose (localized/fixed cutaneous form); (**C**) multiple skin lesions partially covered by hematic crusts on the cephalic region and the left forelimb (disseminated cutaneous form); and (**D**) ascending nodular lymphangitis on the left forelimb (lymphocutaneous form).

**Figure 2 jof-08-01207-f002:**
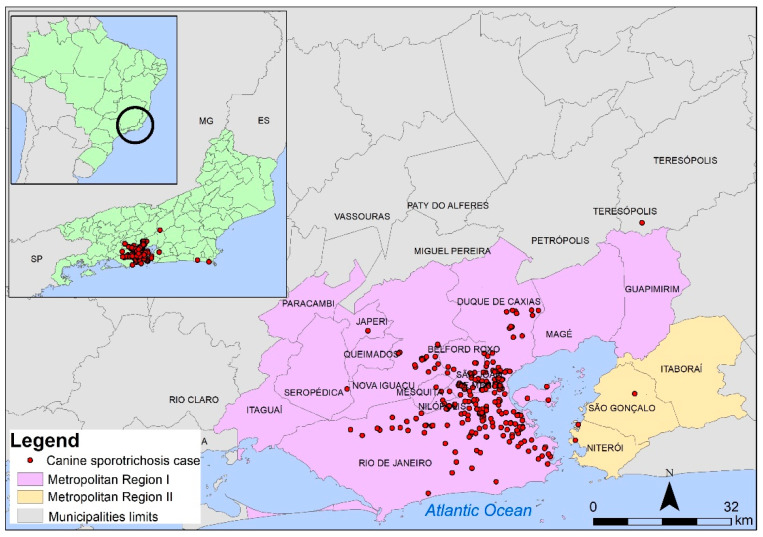
Distribution of cases of canine sporotrichosis in the state of Rio de Janeiro, Brazil, between 1998 and 2018. The limits of the municipalities are shown as black lines. Red dots indicate cases of canine sporotrichosis.

**Figure 3 jof-08-01207-f003:**
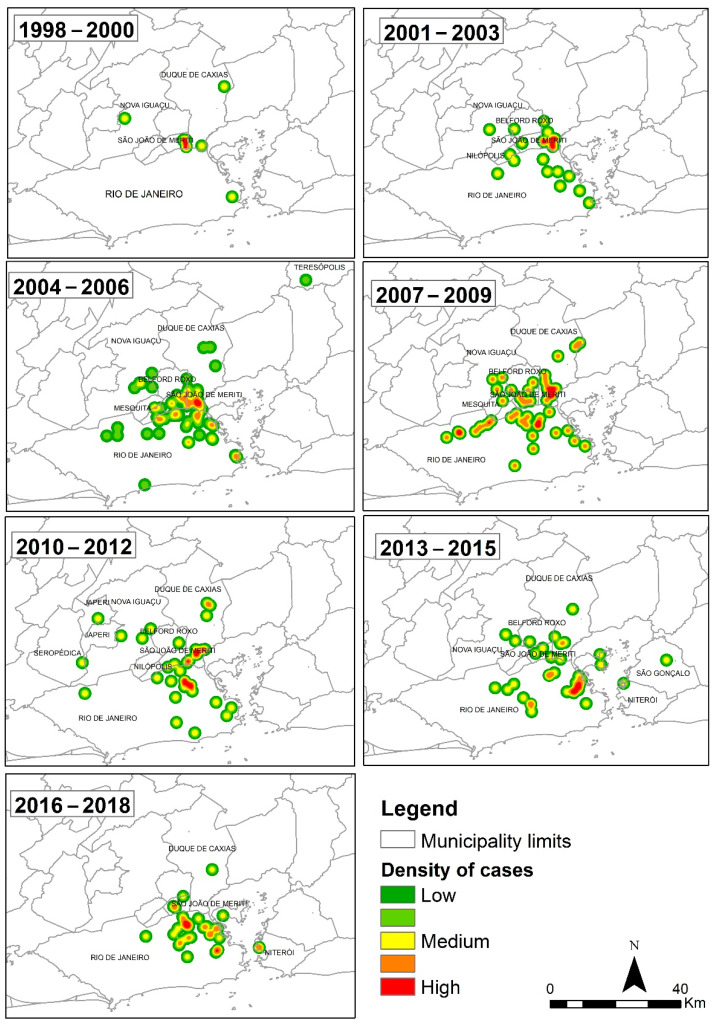
Kernel maps of the 249 canine sporotrichosis cases in the state of Rio de Janeiro, Brazil, over the periods studied.

**Figure 4 jof-08-01207-f004:**
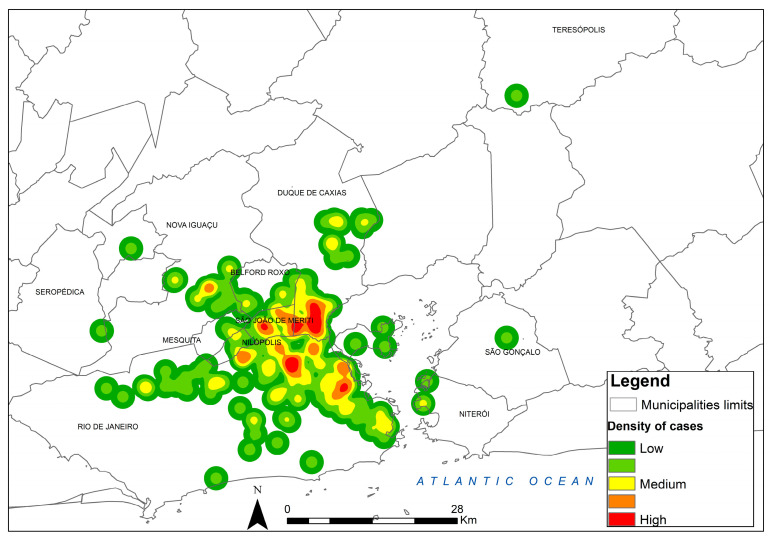
Kernel map of canine sporotrichosis cases in the state of Rio de Janeiro, Brazil, between 1998–2018. Source: Instituto Nacional de Infectologia Evandro Chagas/Fiocruz.

**Figure 5 jof-08-01207-f005:**
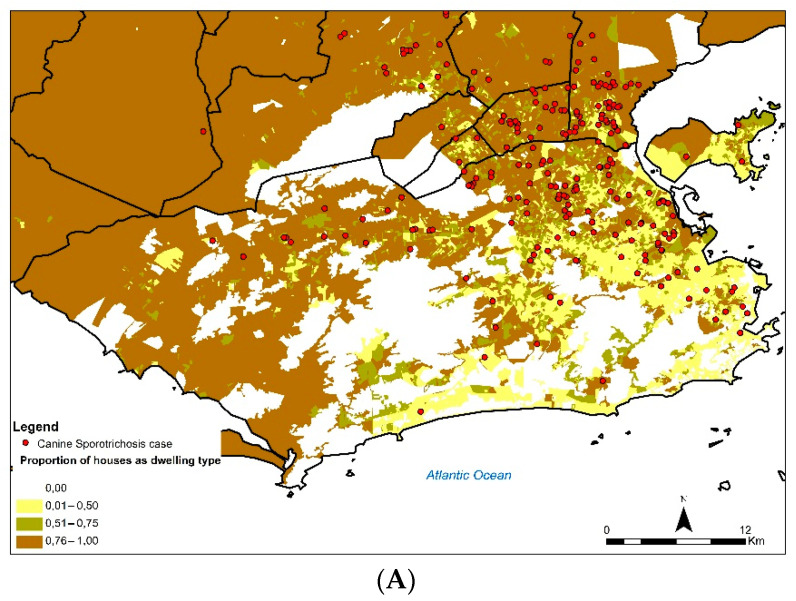
Number of cases of canine sporotrichosis in Rio de Janeiro, Brazil, between 1998 and 2018, superimposed on the proportion of the following indicators: (**A**) Proportion of houses as predominant dwelling type; (**B**) literate responsible persons; (**C**) households with garbage thrown in the river, lake, or sea; and (**D**) households without paving.

**Table 1 jof-08-01207-t001:** Distribution of canine sporotrichosis cases by municipality of residence in the state of Rio de Janeiro, Brazil, 1998–2018.

Region/Municipality of Residence	n	Relative (%)	1998–2000	2001–2003	2004–2006	Period 2007–2009	2010–2012	2013–2015	2016–2018
MR I									
Rio de Janeiro	152	54.28	1	10	35	28	24	26	28
Duque de Caxias	64	22.86	2	9	21	13	10	6	3
São João de Meriti	26	9.28	4	0	8	5	3	3	3
Nova Iguaçu	15	5.36	1	1	5	3	3	2	0
Belford Roxo	5	1.78	0	1	1	1	1	1	0
Nilópolis	4	1.43	0	1	3	0	0	0	0
Magé	2	0.71	0	0	1	1	0	0	0
Mesquita	1	0.36	0	0	0	1	0	0	0
Seropédica	1	0.36	0	0	0	0	1	0	0
Japeri	1	0.36	0	0	0	0	1	0	0
Guapimirim	1	0.36	0	0	0	0	0	1	0
MR II									
São Gonçalo	2	0.71	0	0	0	0	1	1	0
Niterói	2	0.71	0	0	0	0	0	0	2
Itaboraí	1	0.36	0	0	0	0	1	0	0
Coastal Shores region									
Araruama	1	0.36	0	0	0	0	0	0	1
Arraial do Cabo	1	0.36	0	0	0	0	0	0	1
Mountain region									
Teresópolis	1	0.36	0	0	1	0	0	0	0
Rio de Janeiro State	280 *	100.00	8	22	75	52	45	40	38

* Among the 295 canine cases, the municipality of residence was missing in 15 cases. MR: Metropolitan region.

**Table 2 jof-08-01207-t002:** Statistical summary of the classification of each kernel density according to socioeconomic indicator.

Variables	Low (Average [Range])	Medium (Average [Range])	High (Average [Range])
P_DOM_CAS (Dwelling type)	0.728 [0.000–1.000]	0.737 [0.086–1.000]	0.734 [0.000–1.000]
P_R_ALF (Literate)	0.965 [0.880–1.000]	0.954 [0.874–0.997]	0.965 [0.796–1.000]
P_DOM_ESG (Sewage)	0.044 [0.000–0.697]	0.048 [0.000–0.594]	0.028 [0.000–0.703]
P_DOM_LIX (Garbage in water)	0 [0.000–0.697]	0 [0.000–0.000]	0.001 [0.000–0.034]
P_R_1SM (Monthly income)	0.357 [0.000–0.833]	0.409 [0.089–0.778]	0.373 [0.036–0.777]
P_DOM_SPA * (No paving)	0.166 [0.000–1.000]	0.059 [0.000–0.523]	0.056 [0.000–0.980]
P_DOM_LLO (Street garbage)	0.028 [0.000–0.395]	0.034 [0.000–0.448]	0.061 [0.000–1.000]

Legend: P_DOM_CAS = proportion of houses as dwelling type; P_R_ALF = proportion of literate responsible persons; P_DOM_ESG = proportion of households with sewage via a rudimentary cesspool or ditch; P_DOM_LIX = proportion of households with garbage thrown in the river, lake, or sea; P_R_1SM = proportion of responsible persons with a monthly income of up to one minimum wage; P_DOM_SPA = proportion of households without paving; and P_DOM_LLO = proportion of households with garbage accumulated in the street. * Deemed significant via the nonparametric Kruskal–Wallis test (alpha < 0.05).

## Data Availability

Not applicable.

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
