# Peer review of "Spatial Distribution of Canine Sporotrichosis in Rio de Janeiro, Brazil (1998–2018) and Its Correlation with Socioeconomic Conditions"

_jof, 2022, doi:10.3390/jof8111207_

Round 1
Reviewer 1 Report
This manuscript, number jof-1991803, entitled: “Spatial distribution of canine sporotrichosis in Rio de Janeiro, Brazil (1998 – 2018) and its correlation with socioeconomic conditions” by Figueiredo et al., describes the occurrence of canine sporotrichosis in the state of Rio de Janeiro, Brazil, from 1998 to 2018 and its correlation with socioeconomic characteristics using exploratory spatial data analysis. The authors conclude that incidence of cases of sporotrichosis in dogs increases as socioeconomic status decreases but not necessarily linked to extreme poverty.
Major comments:
1. Please in Introduction and Discussion sections, include the reference Pereira GM et al., 2021 (https://doi.org/10.29374/2527-2179.bjvm002421).
2. Please in the Material and Methods section include information of the diagnostic method used, clinical presentation, and fungus species identified.
3. Please in the Discussion section include information about the species of Sporothrix mostly reported in cases of sporotrichosis in dogs in Brazil and other regions of the world.
4. Please improve the Conclusions section: be more specific with the findings.
5. Please improve the quality of the figures. Reduce the compression factor of JPEG (as possible by the maximum file size standard) for sharper images (when zooming image, there are some fuzzy areas).
6. In Table 2, to reduce remembering variable abbreviations, consider to including a new column (between “Variables” and “Low”) with one to three natural words (not the detailed description of the legend) as a healthy redundancy for not jumping between the table and the legend. Examples: Dwelling type, responsible, garbage in water, street garbage, etc.
7. In Table 2, to ease finding the variable, put variable legend in a list.
Minor comments:
1. Page 1, line 40, include comma before S. globose.
2. Page 2, line 40, include comma before dogs.
3. Page 9, line 230, delete trailing period “… legend:” instead of “legend:.”
4. Page 10, line 256, S. brasiliensis in italics (S. brasiliensis)
5. Page 10, line 294, Sporothrix in italics (Sporothrix sp)
6. Page 11, line 336, Sporothrix in italics (Sporothrix spp)
Reviewer 2 Report
There is no attempt to undermine the effort of the team of researchers and authors for this time-assuming project, but this is simply a study that shows the geographic address of households of dog owners whose dog was diagnosed with sporotrichosis over a 20-year period. Unfortunately, the study did not show any meaningful correlations to the dog owners’ demographics using the selected data points. In contrast, the only statistical significance was the negative correlation with the proportion of households without paving!
My suggestions for modifications to the study and the manuscript are as follows:
1- It would be helpful if the research could correlate the concentration of canines diagnosed with the disease to the dog ownership database of the region if exists. In many countries, canines are registered and have licenses. Also, there is no attempt to correlate the study to the feline population in the same concentration of findings perhaps using the same pet ownership database if exists.
2- It would be useful if the manuscript indicates the modalities of the definitive diagnosis that was utilized at these centers such as histopathology, culture, etc., and the sensitivity and specificity of tests that were used.
3- It would be helpful to update the manuscript by identifying the location of the reference laboratories and the animal clinics/animal hospitals where these cases were collected on the map. This is to assure that the concentration of cases has nothing to do with proximity to these collecting and laboratory centers.
4- The manuscript also would benefit from adding a map to describe MR I and MR II in the method section. This will help a reader not familiar with Brazil to understand the study better.
5- It would be helpful if the authors could explain how they eliminated duplications over the 20 years of the study period. In case same canines were seen in several clinics due to the severity of the illness or lack of response to therapy on the first visit or reinfection.
6- Figure 4 map A: It would be useful if authors could change the “Proportion of house type households” to “Proportion of houses as dwelling type” in the map legend (inside the map picture).
Author Response
Reviewer #2:
Firstly, we would like to thank you for all the comments and suggestions.
1- It would be helpful if the research could correlate the concentration of canines diagnosed with the disease to the dog ownership database of the region if exists. In many countries, canines are registered and have licenses. Also, there is no attempt to correlate the study to the feline population in the same concentration of findings perhaps using the same pet ownership database if exists.
Unfortunately, there are no databases with the total number of dogs in the studied region. A similar study on feline sporotrichosis cases diagnosed at INI/Fiocruz is currently being carried out, but the data is still being analyzed (the number of diagnosed feline cases is much higher than dogs).
2- It would be useful if the manuscript indicates the modalities of the definitive diagnosis that was utilized at these centers such as histopathology, culture, etc., and the sensitivity and specificity of tests that were used.
All dogs included in this study were diagnosed through isolation and identification of Sporothrix sp. in appropriate culture media (Schubach et al, 2006), which is the reference method for the diagnosis of sporotrichosis. The above information was included in the page 3, line 96.
3- It would be helpful to update the manuscript by identifying the location of the reference laboratories and the animal clinics/animal hospitals where these cases were collected on the map. This is to assure that the concentration of cases has nothing to do with proximity to these collecting and laboratory centers.
All canine cases in this study were diagnosed at the same institution (INI/Fiocruz). This was even discussed as one of the limitations of the study, as shown below:
“A limitation of this study was that cases diagnosed at other public or private institutions and veterinary clinics could not be included. It is therefore difficult to comprehensively assess the occurrence and distribution of canine sporotrichosis in the state of Rio de Janeiro.
Although our study does not cover other public sporotrichosis care units or private practices, this demand was not so expressive in other locations, especially between 1998 and 2013. It is also observed that, over the years, INI/Fiocruz become a referral center for the cases from the other institutions. Furthermore, since 2014, free-of-charge diagnosis and treatment of animals have become available in other two public institutions in the city of Rio de Janeiro. Until then, INI/Fiocruz was responsible for most reports of animal disease in the state.” (Page 13).
4- The manuscript also would benefit from adding a map to describe MR I and MR II in the method section. This will help a reader not familiar with Brazil to understand the study better.
The map has been added to the manuscript.
5- It would be helpful if the authors could explain how they eliminated duplications over the 20 years of the study period. In case same canines were seen in several clinics due to the severity of the illness or lack of response to therapy on the first visit or reinfection.
All cases included in this study were diagnosed and treated at INI/Fiocruz.
The following text was included in page 3, line 94:
“Each animal had an individual medical record number provided at the first visit. This registration number was valid for all subsequent consultations at INI/Fiocruz throughout the patient's life.”
6- Figure 4 map A: It would be useful if authors could change the “Proportion of house type households” to “Proportion of houses as dwelling type” in the map legend (inside the map picture).
The change was made.
Please see the attachment

Round 2
Reviewer 1 Report
I have no more observations.
Reviewer 2 Report
Thank you for updating the manuscript, edits, and adding maps.
1- Focusing on the previous comment: It would be helpful if the research could correlate the concentration of canines diagnosed with the disease to the dog ownership database of the region if exists. In many countries, canines are registered and have licenses. Also, there is no attempt to correlate the study to the feline population in the same concentration of findings perhaps using the same pet ownership database if exists.
"Unfortunately, there are no databases with the total number of dogs in the studied region."
I suggest the above response be added to the manuscript's discussion as a limitation.
"A similar study on feline sporotrichosis cases diagnosed at INI/Fiocruz is currently being carried out, but the data is still being analyzed."
I also suggest above response be added to either to introduction or discussion.
2- Keywords: " Cat. Sporotrichosis. Epidemiology. Spatial analysis" shouldn't the word "Cat" be replaced with "Canine"? or be added?
3- Conclusions: The canine cases occurred in areas where most of the dwellings are houses and showed a concomitant spatial expansion with human cases.". How did the authors come up with a correlation with human cases by conducting this study?